# A Group-Theoretic Framework for Knowledge Graph Embedding

## Abstract

We have rigorously proved the existence of a group algebraic structure hidden in relational knowledge embedding problems, which suggests that a group-based embedding framework is essential for designing embedding models. Our theoretical analysis explores merely the intrinsic property of the embedding problem itself hence is model independent. Using the proposed framework, one could construct embedding models that naturally accommodate all possible local graph patterns, which are necessary for reproducing a complete graph from knowledge triplets. We reconstruct many state-of-the-art models from the framework and re-interpret them as embeddings with different groups. Moreover, we also propose new instantiation models using simple continuous non-abelian groups.

## 1 Introduction

Knowledge graphs (KGs) are prominent structured knowledge bases for many downstream semantic tasks (Hao et al., 2017). A KG contains an entity set $\mathcal{E} = \{e_i\}$, which correspond to vertices in the graph, and a relation set $\mathcal{R} = \{r_k\}$, which forms edges. The entity and relation sets form a collection of factual triplets, each of which has the form $(\mathbf{e}_i, \mathbf{r}_k, \mathbf{e}_j)$ where $\mathbf{r}_k$ is the relation between the head entity $\mathbf{e}_i$ and the tail entity $\mathbf{e}_j$. Since large scale KGs are usually incomplete due to missing links (relations) amongst entities, an increasing amount of recent works (Bordes et al., 2013; Yang et al., 2014; Trouillon et al., 2016; Lin et al., 2015) have devoted to the graph completion (i.e., link prediction) problem by exploring a low-dimensional representation of entities and relations.

More formally, each relation $\mathbf{r}$ acts as a mapping $\mathbf{O}_\mathbf{r}[\cdot]$ from its head entity $\mathbf{e}_1$ to its tail entity $\mathbf{e}_2$:

$$\mathbf{r} : \mathbf{e}_1 \mapsto \mathbf{O}_\mathbf{r}[\mathbf{e}_1] =: \mathbf{e}_2. \tag{1}$$

The original KG dataset represents these mappings in a tabular form, and the task of KG embedding is to find a better representation for these abstract mappings. For example, in the TransE model (Bordes et al., 2013), relations and entities are embedded in the same vector space, and the operation $\mathbf{O}_\mathbf{r}[\cdot]$ is simply a vector summation: $\mathbf{O}_\mathbf{r}[\mathbf{e}] = \mathbf{e} + \mathbf{r}$. In general, the operation could be either linear or nonlinear, either pre-defined or learned.

Since the entity and relation sets in KGs play conceptually different roles, it is reasonable to represent them differently, which demands a more proper operation $\mathbf{O}_\mathbf{r}[\cdot]$ for bridging the entity and relation sets. On the other hand, the graph completion task relies on the fact that relations are not independent. For example, the hypernym and hyponym are inverse to each other; while kinship relations usually support mutual inferences. Previous studies (Sun et al., 2019; Xu & Li, 2019) have concerned some specific cases of inter-relation dependencies, including (anti-)symmetry and compositional relations. However, one may ask a more general question: could we establish certain fundamental principles to guide the operation design, such that any inter-relation dependency could be naturally captured? In this work, we find the answer is positive, and we term these principles as *hyper-relations*. Group theory emerges as a natural description tool that satisfies the need for finding better operations and *hyper-relations*. In the following sections, we demonstrate the emergence of group definition, namely ***closure, identity, inverse, and associativity***, from a study of knowledge graph itself, and explain in detail an implementation of group theory to tackle general relational embedding problems.

One main contribution of this work is it provides a framework for addressing the KG embedding problem from a novel and more rigorous perspective: the group-theoretic perspective. This frame-

work arises from our investigation of the graph reconstruction problem. We prove that the intrinsic structure of this task *automatically produces* the complete definition of groups. To our best knowledge, this is the first proof that rigorously legitimates the application of group theory in KG embedding. With this framework, we reconstructed several existing models using group theory language (see Sec 3.3), including: TransE (Bordes et al., 2013), TransR (Lin et al., 2015), TorusE (Ebisu & Ichise, 2018), RotatE (Sun et al., 2019), ComplEx (Trouillon et al., 2016), DisMult (Yang et al., 2014).

## 2 RELATED WORKS

From the group theory perspective, our work may be related to the TorusE (Ebisu & Ichise, 2018), the RotatE (Sun et al., 2019) and DihEdral (Xu & Li, 2019) models. The TorusE model tries to frame the KG embedding in a Lie-group and deals with the *compactness problem*. Its authors proved that the additive nature of the operation in the TransE model contradicts the entity regularization. However, if a non-compact embedding space is used, entities must be regularized to prevent the divergence of negative scores. Therefore the TorusE model used $n$-torus, a compact manifold, as the embedding space. In other words, a group is regarded as a manifold, rather than as a set of operations where algebraic structures are more emphasized. Besides, the compactness issue was solved in many later works by bounding the training objective using *Softplus* function (Sun et al., 2019). In the DihEdral model, the $D_4$ group the author used plays the same role as in our work: depicting the transformation of entities by assigning each relation a group element. The motivation of DihEdral is to resolve the non-abelian composition (i.e., the compositional relation formed by $r_1$ and $r_2$ would change if the two are switched). Nevertheless, DihEdral applies a discrete group $D_8$ for relation embedding while using a continuous entity embedding space, which may suffer two problems as discussed in the later Section 3.3. The RotatE model was designed to accommodate symmetric, inversion, and (abelian) composition of triplets at the same time.

Different from those previous works, this work does not target at one or few specific cases but aims at answering the more general question: finding the generative principle of all possible cases, and thus to provide guidance for model designs that can accommodate all cases. More importantly, compared with most preceding works connecting to groups (Xu & Li, 2019; Cai, 2019), the analysis in our current work does not focus on *impose* an implementation of groups. Rather, we start merely by studying the graph reconstruction problem and prove that the intrinsic structure of this task itself *automatically produces* the complete definition of groups. To our the best of our knowledge, this is the first proof that rigorously legitimates the application of group theory in KG embeddings.

## 3 GROUP THEORY IN RELATIONAL EMBEDDINGS

In this section, we formulate the group-theoretic analysis for relational embedding problems. For simplicity, our discussion would start from 1-to-1 mappings and discuss the generalization to more complicated cases in the Appendix D.

Firstly, as most embedding models represent objects (including entities and relations) as vectors, the task of operation design thus can be stated as finding proper transformations of vectors. Secondly, as we mentioned in the introduction, our ultimate goal of reproducing the whole knowledge graph using atomic triplets further requires certain types of local patterns to be accommodated. We now discuss these structures, which in the end naturally leads to the definition of groups in mathematics.

### 3.1 HYPER-RELATION PATTERNS: RELATION-OF-RELATIONS

One difficulty of generating the whole knowledge graph from atomic triplets lies in the fact that different relations are not independent of each other. The task of relation inference relies exactly on their mutual dependency. In other words, there exist certain *relation of relations* in the graph structure, which we term as *hyper-relation patterns*. A proper relation embedding method and the associated operations should be able to capture these hyper-relations.

Now instead of studying exampling cases one by one, we ask the most general question: what are the most fundamental hyper-relations? The answer is quite simple and only contains two types, namely, *inversion* and *composition*:

- **Inversion**: given a relation $\mathbf{r}$, there *may exist* an inversion $\bar{\mathbf{r}}$, such that:

$$\mathbf{r} : \mathbf{e}_1 \mapsto \mathbf{e}_2 \quad \longrightarrow \quad \bar{\mathbf{r}} : \mathbf{e}_2 \mapsto \mathbf{e}_1, \qquad \forall \mathbf{e}_1, \mathbf{e}_2 \in \mathcal{E}. \tag{2}$$

  The inversion captures any relation path with a length equal to 1 (in the unit of relations).

- **Composition**: given two relations $\mathbf{r}_1$ and $\mathbf{r}_2$, there *may exist* a third relation $\mathbf{r}_3$, such that:

$$\left\{ \begin{array}{l} \mathbf{r}_1 : \mathbf{e}_1 \mapsto \mathbf{e}_2 \\ \mathbf{r}_2 : \mathbf{e}_2 \mapsto \mathbf{e}_3 \end{array} \right. \quad \longrightarrow \quad \mathbf{r}_3 : \mathbf{e}_1 \mapsto \mathbf{e}_3, \qquad \forall \mathbf{e}_1, \mathbf{e}_2, \mathbf{e}_3 \in \mathcal{E}. \tag{3}$$

  Any relation paths longer than 1 can be captured by a sequence of compositions.

One may notice the phrase *may exist* in the above definition, this simply emphasizes that the existence of these derived conceptual relations $\bar{\mathbf{r}}$ and $\mathbf{r}_3$ depends on the **specific KG dataset**; while, on the other hand, to accommodate general KG datasets, the **embedding space** should always contains the mathematical representations of these conceptual relations.

An important feature of KG is that with the above two hyper-relations, one could generate any local graph pattern and eventually the whole graph, as relational paths with arbitrary length have been captured. Note the term of *inversion* and *composition* might have different meanings from ones in other works: most existing works study triplets to analyze hyper relations, while the definition we provide above is based purely on relations. This is more general in the sense that any conclusion derived would not depend on entities at all, and some different hyper relations could, therefore, be summarized as a single one. For example, there are enormous discussions on *symmetric* triplets and *anti-symmetric* triplets (Sun et al., 2019), which are defined as:

$$\begin{array}{llll} \textbf{symmetric}: & (\mathbf{e}_1, \mathbf{r}, \mathbf{e}_2) & \longrightarrow & (\mathbf{e}_2, \mathbf{r}, \mathbf{e}_1), \\ \textbf{anti-symmetric}: & (\mathbf{e}_1, \mathbf{r}, \mathbf{e}_2) & \longrightarrow & \neg(\mathbf{e}_2, \mathbf{r}, \mathbf{e}_1). \end{array} \tag{4}$$

In fact, if for any choice of $\mathbf{e}_{1,2}$, one could produce a symmetric pair of true triplets using $\mathbf{r}$, this would imply a property of $\mathbf{r}$ itself, and in which case, one could then simply derive:

$$\bar{\mathbf{r}} = \mathbf{r}. \tag{5}$$

This is a special case of the inversion hyper-relation; and similarly, the anti-symmetric case simply implies $\bar{\mathbf{r}} \neq \mathbf{r}$, which is quite common, and does not require extra design. The deep reason for discussing hyper-relations which relies merely on relations rather than triplets is that the logic of relation inference problem itself is not entity-dependent.

### 3.2 EMERGENT GROUP THEORY

To accommodate both general inversions and general compositions, we now derive explicit requirements on the relation embedding model. We start by defining the *product* of two relations $\mathbf{r}_1$ and $\mathbf{r}_2$: $\mathbf{r}_1 \cdot \mathbf{r}_2$, as subsequently "finding the tail" twice according to the two relations, i.e.

$$\mathbf{O}_{\mathbf{r}_1 \cdot \mathbf{r}_2} \big[\, \cdot \,\big] := \mathbf{O}_{\mathbf{r}_1} \big[ \mathbf{O}_{\mathbf{r}_2}[\cdot] \big]. \tag{6}$$

With the above definition, equation 3 can be rewritten as: $\mathbf{r}_3 = \mathbf{r}_1 \cdot \mathbf{r}_2$. One would realize that the following properties should be supported by a proper embedding model:

1. **Inverse element**: to allow the possible existence of inversion, the elements $\bar{\mathbf{r}}$ should also be an element living in the same relation-embedding space[1].

2. **Closure**: to allow the possible existence of composition, in general, the elements $\mathbf{r}_1 \cdot \mathbf{r}_2$ should also be an element living in the same relation-embedding space[2].

3. **Identity element**: the possibly existing inverse and composition together define another special and unique relation:

$$\mathbf{i} = \mathbf{r} \cdot \bar{\mathbf{r}}, \qquad \forall \mathbf{r} \in \mathcal{R}. \tag{7}$$

  This element should map any entity to itself, and thus we call it *identity element*.

---

[1]Given a graph, not all inversions correspond to meaningful relations, but an embedding model should be able to capture this possibility in general.

[2]Given a graph, not all compositions correspond to meaningful relations, but an embedding model should be able to capture this possibility in general.

4. **Associativity**: In a relational path with the length longer than three (containing three or more relations $\{r_1, r_2, r_3, ...\}$), as long as the sequential order does not change, the following two compositions should produce the same result:

$$(\mathbf{r}_1 \cdot \mathbf{r}_2) \cdot \mathbf{r}_3 = \mathbf{r}_1 \cdot (\mathbf{r}_2 \cdot \mathbf{r}_3). \tag{8}$$

See Appendix A for the proof.

5. **Commutativity/Nonconmmutativity**: In general, commuting two relations in a composition, i.e. $\mathbf{r}_1 \cdot \mathbf{r}_2 \leftrightarrow \mathbf{r}_2 \cdot \mathbf{r}_1$, may compose either the same or different results. In real graphs, any cases may exist, and a proper embedding method should be able to accommodate both.[3]

The first four properties are exactly the definition of a ***group***. In other words, *the group theory automatically emerges from the relational embedding problem itself, rather than being applied manually.* This is quite convincing evidence that group theory is indeed the most natural language for relational embeddings if one aims at ultimately reproducing all possible local patterns in graphs. Besides, the fifth property on *commutativity/nonconmmutativity* are actually termed as **abelian/nonabelian** in the group theory language. Since abelian is more special, to accommodate both possibilities, one should, in general, consider a **nonabelian group for the relation embedding**, and guarantee at the same time it contains at least one nontrivial abelian subgroup.

More explicitly, given a graph, to implement a group structure in embedding, one should embed all relations as group elements, which are parametrized by certain group parameters. For instance: the translation group $T$ can be parametrized by a real number $\delta$. And correspondingly, due to its vector nature, the embedding of entities could be regarded as a **representation (rep)** space of the same group. For the translation group, $\mathbb{R}$ (the real field) is a rep space of $T$.

This suggests the group representation theory is useful in knowledge graph embedding problems when talking about entity embeddings, and we leave this as a separate topic for subsequent works later. In the later section, we provide a general recipe for the graph embedding implementation.

### 3.3 EMBEDDING MODELS USING DIFFERENT GROUPS

In this section, we discuss embedding methods using different groups, from simple ones as $T$ (the translation group) and $U(1)$, to complicated ones including $SU(2)$, $GL(n, \mathbb{V})$ (where $\mathbb{V}$ could be any type of fields), or even $Aff(\mathbb{V})$. It is important to note that, in practice, **continuous groups** are more reasonable than discrete ones, due to the two following reasons:

- The entity embedding space is usually continuous, which matches reps of the continuous group better. If used to accommodate a discrete group, a continuous space always contains infinite copies of irreducible reps of that group, which makes the analysis much more difficult.
- When training the embedding models, a gradient-based optimization search would be applied in the parameter space. However, different from continuous groups whose group parameter are also continuous, the parametrization of a discrete group uses discrete values, which brings in extra challenges for the training procedure.

With the two reasons above, we thus mainly consider continuous groups which are more reasonable choices. Although for completeness, we also compare the group $D_4$ as it is implemented in DihEdral (Xu & Li, 2019). Two other important feature of a group are compactness and commutativity, which we would mention in each case below.

Besides the relational embedding group $\mathcal{G}$, the entity embedding space and the similarity measure also need to be determined. As discussed above, the entity embedding should be a proper rep space of $\mathcal{G}$. While for similarity measure $d(\cdot)$, we choose among the popular ones including $L_p$-norms ($L_p$) and the cos-similarity (cos). We would notice many choices reproduce some precedent works.

We summarize the results of several chosen examples in Table 1, and put a thorough discussion in Appendix D. Note in the Table 1, some groups have not been studied, but there are still some existing models which use a quite similar embedding space. The major gap, between the existing models and their group embedding counterparts, is the constraint from group structures on the parametrization. which is discussed in details in Appendix D.

---

[3]See Appendix B for a more detailed explanation and practical examples.

| Group | Space | Commutativity | $d(\cdot)$ | Studied | Related work |
|---|---|---|---|---|---|
| $T$ | $\mathbb{R}^n$ | abelian | $L_p$ | ✓ | TransE (Bordes et al., 2013) |
| $U(1)$ | $\mathbb{C}^n$ | abelian | $L_p$ | ✓ | RotatE (Sun et al., 2019) |
| $U(1)$ | $\mathbb{T}^n$ | abelian | $L_p$ | ✓ | TorusE (Ebisu & Ichise, 2018) |
| $SO(3)$ | $\mathbb{R}^{3n}$ | nonabelian | $L_p$ | – | – |
| $SU(2)$ | $\mathbb{C}^{2n}$ | nonabelian | $L_p$ | – | – |
| $GL(1, \mathbb{R})$ | $\mathbb{R}^n$ | abelian | cos | ✓ | DisMult (Yang et al., 2014) |
| $GL(1, \mathbb{C})$ | $\mathbb{C}^n$ | abelian | cos | ✓ | ComplEx (Trouillon et al., 2016) |
| $GL(n, \mathbb{R})$ | $\mathbb{R}^n$ | nonabelian | cos | – | RESCAL (Nickel et al., 2011) |
| $Aff(\mathbb{R}^n)$ | $\mathbb{R}^n$ | nonabelian | $L_p$ | – | TransR (Lin et al., 2015) |
| $D_4$ | $\mathbb{R}^n$ | nonabelian | $L_p$ | ✓ | DihEdral (Xu & Li, 2019) |

Table 1: Examples of the group embedding.

# 4 GROUP EMBEDDING FOR KNOWLEDGE GRAPHS

In this section, we would firstly provide a general recipe for the group embedding implementation, and then provide an explicit example, which applies a continuous nonabelian group that has not been studied in any precedent works before.

## 4.1 A GENERAL GROUP EMBEDDING RECIPE

We summarize the group embedding procedure as following:

1. Given a graph, choose a proper group $\mathcal{G}$ for embedding. The choice may concern property of the task, such as commutativity and so on. And as stated above, in most general cases, a nonabelian continuous group should be proper.

2. Choose a rep-space for the entity embedding. For simplicity, one could use multiple ($n$) copies of the same rep $\rho$, which is the case of most existing works. Suppose $\rho$ is a $p$-dim rep, then the total dimension of entity embedding would be $pn$, which is written as a vector $\vec{v}_e$. Roughly speaking, $k$ captures the relational structure and $n$ encodes other feature.

3. Choose a proper parametrization of $\mathcal{G}$, that is, choose a set of parameters indexing all group elements in $\mathcal{G}$. Suppose the number of parameters required to specify a group element is $q$, then the total dimension of relation embedding $\vec{v}_r$ would be $qn$. A group element can now be expressed as a block-diagonal matrix $\mathbf{R}_r$, with each block $\mathbf{M}_i$ being a $p \times p$ matrix whose entries are determined by the vector $\vec{v}_r$.

4. Choose a similarity measure $d[\cdot]$, the score value of a triplet $(\mathbf{e}_1, \mathbf{r}, \mathbf{e}_2)$ is then:

$$d\big[\mathbf{R}_r \cdot \vec{v}_{e_1}, \ \ \vec{v}_{e_2}\big] \tag{9}$$

## 4.2 AN EXAMPLE OF NONABELIAN GROUP EMBEDDING: **SU2E**

The 2D special unitary group $SU(2)$ is one of the simplest continuous nonabelian group. As an illustrative demonstration, we construct an embedding model with $SU(2)$ structure and implement it in real experiments. As we mentioned in Table 1, $SO(3)$ is also a nice example group. The reason we study $SU(2)$ rather than $SO(3)$, is from a well-known mathematical fact[4]: the universal cover of $SO(3)$ is **Spin**(3) which is isomorphic to $SU(2)$; or in the representation language, any rep of $SO(3)$ corresponds to an integer rep of $SU(2)$. This reveals that a study of $SU(2)$ is more general and could easily cover the theory of $SO(3)$.

Following the general recipe above, after determining the group $\mathcal{G} = SU(2)$, we choose a proper rep-space for entity embedding: $[\mathbb{C}^2]^{\otimes n}$, which consists $n$-copies of $\mathbb{C}^2$. Each $\mathbb{C}^2$ subspace transforms as the standard rep-space of $SU(2)$. All relations thus act as $2n \times 2n$ block diagonal matrix, with each block being a $2 \times 2$ complex matrix carrying the standard representation of $SU(2)$.

---

[4]https://en.wikipedia.org/wiki/3D_rotation_group

Next, we choose a proper parametrization of $SU(2)$. An analysis with the corresponding Lie algebra $\mathfrak{su}(2)$ shows that any group element could be written as (Hall & Hall, 2003):

$$e^{i\alpha[\hat{\mathbf{n}}\cdot\vec{\mathbf{J}}]} = \cos\alpha\hat{1} + i\sin\alpha\hat{\mathbf{n}}\cdot\vec{\mathbf{J}}, \tag{10}$$

where $\alpha$ is a rotation angle taken from $[0, 2\pi]$, and $\hat{\mathbf{n}}$ is a unit vector on 2-sphere, represented by two other angles $(\theta, \phi)$; moreover, the symbol $\hat{1}$ means an identity matrix, and $\vec{\mathbf{J}}$ are three generators of the group $SU(2)$: $(\mathbf{J}_x, \mathbf{J}_y, \mathbf{J}_z)$, which, in the standard rep have the following form:

$$J_x = \begin{bmatrix} 0 & 1 \\ 1 & 0 \end{bmatrix}, \qquad J_y = \begin{bmatrix} 0 & \text{-}i \\ i & 0 \end{bmatrix}, \qquad J_z = \begin{bmatrix} 1 & 0 \\ 0 & \text{-}1 \end{bmatrix}. \tag{11}$$

Put all together, our group embedding is then fixed as:

$$\begin{aligned} \mathbf{e} &\implies \vec{v}_e = (x_1, y_1, \ x_2, y_2, \ \cdots, \ x_n, y_n), &\forall \mathbf{e} \in \mathcal{E}; \\ \mathbf{r} &\implies \vec{v}_r = (\alpha_1, \hat{\mathbf{n}}_1, \ \alpha_2, \hat{\mathbf{n}}_2, \ \cdots, \ \alpha_n, \hat{\mathbf{n}}_n), &\forall \mathbf{r} \in \mathcal{R}; \end{aligned} \tag{12}$$

where $x_i$ and $y_i$ are complex numbers, and $\alpha_i$ and $\hat{\mathbf{n}}_i = (\theta_i, \phi_i)$ represent angles. In a triplet, the relation $\vec{v}_r$ acts as a block diagonal matrix $\mathbf{R}_r$, with each $2 \times 2$ block $\mathbf{M}_i$ parametrized as (Hall & Hall, 2003):

$$\begin{bmatrix} \cos\alpha_i + i\sin\alpha_i\sin\theta_i & ie^{-i\phi_i}\cdot\sin\alpha_i\cos\theta_i \\ ie^{-i\phi_i}\cdot\sin\alpha_i\cos\theta_i & \cos\alpha_i - i\sin\alpha_i\sin\theta_i \end{bmatrix}$$

An operation of relation $\mathbf{r}$ on $\mathbf{e}$ would act each block matrix $M_i$ in the subspace of $(x_i, y_i)$. This completes the discussion of group embedding. We design the model loss function as follows:

$$L = -\log\sigma(\gamma - d_r(\mathbf{e_1}, \mathbf{e_2})) - \sum_{i=1}^{n} p(\mathbf{e'_{1i}}, r, \mathbf{e'_{2i}})\log\sigma(d_r(\mathbf{e'_{1i}}, \mathbf{e'_{2i}}) - \gamma) \tag{13}$$

$$p(\mathbf{e'_{1j}}, r, \mathbf{e'_{2j}}| \{(\mathbf{e_{1i}}, r_i, \mathbf{e_{2i}})\}) = \frac{\exp\alpha f_r(\mathbf{e'_{1j}}, \mathbf{e'_{2j}})}{\sum_i \exp\alpha f_r(\mathbf{e'_{1i}}, \mathbf{e'_{2i}})} \tag{14}$$

where $\sigma$ is the Sigmoid function, $\gamma$ is the margin used to prevent over-fitting, $d_r$ is the similarity measure, we use the conventional $L_2$-norm. $\mathbf{e'_{1i}}$ and $\mathbf{e'_{2i}}$ are negative samples while $p(\mathbf{e'_{1i}}, r, \mathbf{e'_{2i}})$ is the adversarial sampling mechanism with temperature $\alpha$ we adopt self-adversarial negative sampling setting from (Sun et al., 2019). $f_r$ is a $L_2$-norm score function. We term the resulting model as **SU2E**. For other implementation details, we would mention in the next section.

## 5 EXPERIMENTS

### 5.1 EXPERIMENTAL SETUP

**Datasets:** The most popular public knowledge graph datasets include FB15K (Bollacker et al., 2008) and WN18 (Miller, 1995). FB15K-237 (Toutanova & Chen, 2015) and WN18RR (Dettmers et al., 2018) datasets were derived from these two, in which the inverse relations were removed. FB15K dataset is a huge knowledge base with general facts containing 1.2 billion instances of more than 80 million entities. For benchmarking, usually, a frequency filter was applied to obtain occurrence larger than 100 resulting in 592,213 instances with 14,951 entities and 1,345 relation types. WN18 was extracted from WordNet (Miller, 1995) dictionary and thesaurus, the entities are word senses and the relations are lexical relations between them. It has 151,442 instances with 40,943 entities and 18 relation types.

**Evaluation Protocols:** We use three categories of protocols for evaluations, namely, cut-off Hit ratio (H@N), Mean Rank(MR) and Mean Reciprocal Rank (MRR). H@N measures the ratio of correct entities predictions at a top $n$ prediction result cut-off. Following the baselines used in recent literatures, we chose $n = 1, 3, 10$. MR evaluates the average rank among all the correct entities. MRR is the average rank inverse rank of the correct entities.

**Implementation Details:** We implemented our model using pytorch[5] framework and experimented on a server with an Nvidia Titan-1080 GPU. The Adam (Kingma & Ba, 2014) optimizer was used with the default $\beta_1$ and $\beta_2$ settings. A learning rate scheduler observing validation loss decrease was used to reduce learning rate by half after patience of 3000. Batch-size was set at 1024. We did a grid search on the following hyper-parameters: embedding dimension $d \in \{100, 200, 250\}$; learning rate $\eta \in \{1e-4, 3e-4, 5e-5\}$; number of negative samples during training $n_{neg} \in \{128, 256\}$; adversarial negative sampling temperature $\alpha \in \{0.5, 1.0\}$; loss function margin $\gamma \in \{6, 9, 12, 24, 36, 48\}$.

## 5.2 RESULTS AND MODEL ANALYSIS

Empirical results from FB15k and WN18 are reported in Table 2 and 3. We compared the embedding results of different groups, including $T$, $U(1)$, $GL(1, \mathbb{R})$, $GL(1, \mathbb{C})$, $D_4$, and $SU(2)$, which are categorized by two major properties: the commutativity and the continuity. As discussed in Sec. 3.3, these groups have been implicitly applied in existing state-of-the-art models.

For $SU(2)$, we report the result of our own experiments. Results of the other models are taken from their original literature: TransE using group $T$ was proposed in Bordes et al. (2013); RotatE using group $U(1)$ was proposed in Sun et al. (2019) while TorusE with the same group was proposed in Ebisu & Ichise (2018); group $GL(1, \mathbb{V})$ was implemented in DisMult (Yang et al., 2014) with $\mathbb{V} = \mathbb{R}$ and in ComplEx (Trouillon et al., 2016) with $\mathbb{V} = \mathbb{C}$; and DihEdral with group $D_4$ was proposed in Xu & Li (2019).

| Group | Commutativity | Continuity | MR | MRR | H@1 | H@3 | H@10 | Example |
|---|---|---|---|---|---|---|---|---|
| T | abelian | continuous | - | 0.463 | 0.297 | 0.578 | 0.749 | TransE |
| U(1) | abelian | continuous | **40** | **0.797** | **0.746** | **0.830** | **0.884** | RotatE |
| U(1) | abelian | continuous | - | 0.733 | 0.674 | 0.771 | 0.832 | TorusE |
| GL(1, ℝ) | abelian | continuous | - | 0.654 | 0.546 | 0.733 | 0.824 | DistMult |
| GL(1, ℂ) | abelian | continuous | - | 0.692 | 0.599 | 0.759 | 0.840 | ComplEx |
| $D_4$ | non-abelian | discrete | - | 0.728 | 0.648 | 0.782 | 0.864 | DihEdral |
| SU(2) | non-abelian | continuous | 42 | 0.776 | 0.710 | 0.823 | 0.881 | SU2E |

Table 2: Link prediction results on FB15K dataset

| Group | Commutativity | Continuity | MR | MRR | H@1 | H@3 | H@10 | Example |
|---|---|---|---|---|---|---|---|---|
| T | abelian | continuous | - | 0.495 | 0.113 | 0.888 | 0.943 | TransE |
| U(1) | abelian | continuous | 309 | 0.949 | **0.944** | 0.952 | 0.959 | RotatE |
| U(1) | abelian | continuous | - | 0.947 | 0.943 | 0.950 | 0.954 | TorusE |
| GL(1, ℝ) | abelian | continuous | - | 0.822 | 0.728 | 0.914 | 0.936 | DistMult |
| GL(1, ℂ) | abelian | continuous | - | 0.941 | 0.936 | 0.945 | 0.949 | ComplEx |
| $D_4$ | non-abelian | discrete | - | 0.946 | 0.942 | 0.949 | 0.954 | DihEdral |
| SU(2) | non-abelian | continuous | **207** | **0.950** | **0.944** | **0.954** | **0.960** | SU2E |

Table 3: Link prediction results on WN18 dataset

Results on datasets FB15K-237 and WN18RR are demonstrated in Table 4 and 5 respectively. We remove TorusE from the tables due to the absence of results in the original work, and use results in Nguyen et al. (2017) for TransE.

In the FB15k dataset, the main hyper-relation is anti-/symmetry and inversion. The dataset has a vast amount of unique entities. Shown in Table 2, the RotatE model achieved good performance in this dataset. We should also point out that RotatE model result is achieved on large embedding dimension setting namely 1000 (effective 2000) while we use a setting of only 250 (effective 1000) across all the tasks. SU2E achieved comparable result across the metrics. On the other hand, in FB15k-237 dataset since inversion relations are removed, the dominant portion of hyper-relations becomes the composition. We can see RotatE fail short on non-abelian hyper-relations in this task. Shown in 4, the continuous non-abelian group method SU2E outperformed most of the metrics. The DihEdral model suffers from the problems we mentioned in Section 3.3.

---

[5]https://www.pytorch.org

| Group | Commutativity | Continuity | MR | MRR | H@1 | H@3 | H@10 | Example |
|-------|---------------|------------|-----|------|------|------|-------|---------|
| T | abelian | continuous | 357 | 0.294 | - | - | 0.465 | TransE |
| U(1) | abelian | continuous | 177 | 0.338 | 0.241 | 0.375 | **0.533** | RotatE |
| $GL(1,\mathbb{R})$ | abelian | continuous | - | 0.241 | 0.155 | 0.263 | 0.419 | DistMult |
| $GL(1,\mathbb{C})$ | abelian | continuous | - | 0.247 | 0.158 | 0.275 | 0.428 | ComplEx |
| $D_4$ | non-abelian | discrete | - | 0.300 | 0.204 | 0.332 | 0.496 | DihEdral |
| SU(2) | non-abelian | continuous | **169** | **0.340** | **0.243** | **0.376** | 0.532 | SU2E |

Table 4: Link prediction results on FB15K-237 dataset

| Group | Commutativity | Continuity | MR | MRR | H@1 | H@3 | H@10 | Example |
|-------|---------------|------------|-----|------|------|------|-------|---------|
| T | abelian | continuous | 3384 | 0.226 | - | - | 0.501 | TransE |
| U(1) | abelian | continuous | 3340 | 0.476 | 0.428 | 0.492 | 0.571 | RotatE |
| $GL(1,\mathbb{R})$ | abelian | continuous | - | 0.430 | 0.390 | 0.440 | 0.490 | DistMult |
| $GL(1,\mathbb{C})$ | abelian | continuous | - | 0.440 | 0.410 | 0.460 | 0.510 | ComplEx |
| $D_4$ | non-abelian | discrete | - | **0.486** | **0.442** | **0.505** | 0.557 | DihEdral |
| SU(2) | non-abelian | continuous | **2968** | 0.474 | 0.424 | 0.490 | **0.574** | SU2E |

Table 5: Link prediction results on WN18RR dataset

In the WN18 dataset, the SU2E outperformed all the baselines on all metrics shown in Table 3. The recent model RotatE and DihEdral achieved comparable results. The WN18RR dataset removes the inversion relations from WN18, left only 11 relations and most of them are symmetry patterns. We can see from Table 5, the DihEdral model and SU2E model performed well on this dataset due to their nonabelian nature. Besides, as DihEdral uses a discrete parametrization in relation embedding, it may benefit in special cases where the number of relations is extremely small (11 in this task).

Drawn from the experiments, two factors significantly impact the embedding model performance: the embedding dimension, and group attributes (including commutativity and continuity). Our current study provides the first systematic analysis of the second factor. As theoretically analyzed in Section 3.2, and empirically shown above, continuous nonabelian groups are more reasonable choices for general tasks. It is important to note that the SU2E proposed above plays as an exampling model for our group embedding framework, and it uses the simplest continuous non-abelian group. Much more efforts could be devoted in this direction in the future.

# 6 Conclusion and Future Work

We proved for the first time the emergence of a group definition in the KG representation learning. This proof suggests that relational embeddings should respect the group structure. A novel theoretic framework based on group theory was therefore proposed, termed as the *group embedding* of relational KGs. Embedding models designed based on our proposed framework would automatically accommodate all possible hyper-relations, which are building-blocks of the link prediction task.

From the group-theoretic perspective, we categorize different embedding groups regarding commutativity and the continuity and empirically compared their performance. We also realize that many recent models correspond to embeddings using different groups. Generally speaking, a continuous non-abelian group embedding should be powerful for a generic KG completion task. We demonstrate this idea by examining a simple exampling model: **SU2E**. With $SU(2)$ as the embedding group, it showed promising performance in challenging tasks where hyper-relations become crucial.

In the proposed framework, beside embedding relations as group elements, entity embeddings live in different representation space of the corresponding group. And therefore an investigation of group representation theory in entity embedding is highly demanded. We leave this in future works. On the other hand, although empirical evaluations focus on linear models, it is important to note that the proof of the group structure only relies on the KG task itself. This means our conclusion also works for more general models, including neural-network-based ones. Beyond KG embeddings, the same analysis could be applied to other representation learning where intrinsic relational structures are prominent. An implementation of group structures in more general cases would be very interesting.

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

## A  Proof of hyper-associativity in knowledge graph relations

The associativity is actually rooted in our definition of $\mathbf{r}_1 \cdot \mathbf{r}_2$ in equation 6 through the subsequent operating sequence in the entity space:

$$\mathbf{O}_{\mathbf{r}_1 \cdot \mathbf{r}_2}\big[\,\cdot\,\big] := \mathbf{O}_{\mathbf{r}_1}\big[\mathbf{O}_{\mathbf{r}_2}[\cdot]\big], \tag{15}$$

from which, we can derive directly that:

$$\begin{aligned}
\mathbf{O}_{(\mathbf{r}_1 \cdot \mathbf{r}_2) \cdot \mathbf{r}_3}\big[\,\cdot\,\big] &= \mathbf{O}_{\mathbf{r}_1 \cdot \mathbf{r}_2}\big[\mathbf{O}_{\mathbf{r}_3}[\cdot]\big] \\
&= \mathbf{O}_{\mathbf{r}_1}\big[\mathbf{O}_{\mathbf{r}_2}[\mathbf{O}_{\mathbf{r}_3}[\cdot]]\big] \\
&= \mathbf{O}_{\mathbf{r}_1}\big[\mathbf{O}_{\mathbf{r}_2 \cdot \mathbf{r}_3}[\cdot]\big] = \mathbf{O}_{\mathbf{r}_1 \cdot (\mathbf{r}_2 \cdot \mathbf{r}_3)}\big[\,\cdot\,\big],
\end{aligned} \tag{16}$$

which then leads to the association equation below:

$$(\mathbf{r}_1 \cdot \mathbf{r}_2) \cdot \mathbf{r}_3 = \mathbf{r}_1 \cdot (\mathbf{r}_2 \cdot \mathbf{r}_3). \tag{17}$$

This completes the proof of associativity in knowledge graphs.

## B  An explicit example of hyper-associativity

While the other four properties derived in the main part is easy to comprehend, the associativity may not be intuitive. To help readers understand the practical meaning of associativity in real life cases, here we provide a simple example of the relational hyper-associativity:

$$\mathbf{r}_1 = isBrotherOf, \quad \mathbf{r}_2 = isMotherOf, \quad \mathbf{r}_3 = isFatherOf. \tag{18}$$

Meanwhile, the following composition relations are also meaningful:

$$\mathbf{r}_1 \cdot \mathbf{r}_2 = isUncleOf, \quad \mathbf{r}_2 \cdot \mathbf{r}_3 = isGrandmotherOf. \tag{19}$$

In this example, one could easily see that:

$$(\mathbf{r}_1 \cdot \mathbf{r}_2) \cdot \mathbf{r}_3 = \mathbf{r}_1 \cdot (\mathbf{r}_2 \cdot \mathbf{r}_3) = isGranduncleOf. \tag{20}$$

This completes a simple demonstration of the hyper-associativity.

## C  Examples of non-commutative relations

We provide a simple illustrative examples for non-commutative compositions. We consider the following real world kinship:

$$\mathbf{r}_1 = isMotherOf, \quad \mathbf{r}_2 = isFatherOf. \tag{21}$$

Clearly, the composition $\mathbf{r}_1 \cdot \mathbf{r}_2$ and $\mathbf{r}_2 \cdot \mathbf{r}_1$ correspond to *isGrandmotherOf* and *isGrandfatherOf* relations respectively, which are different. This is a simple example of non-commutative cases.

## D  Detailed discussion of exampling group embeddings

In this appendix, we give a thorough discussion on examples listed in the Table 1. We would explain in details some minor differences between models obtained directly from group embedding theory here, and existing models which may not concern/obey group structures in precedent works.

### D.1  Group: $T$

Use multiple ($n$) copies of $T$, the translation group, for the relation embedding. This is a ***noncompact abelian*** group. The simplest rep-space would be the real field $\mathbb{R}$, which should also appear $n$ times as $\mathbb{R}^n$. The group embedding then produces the following embedding vectors:

$$\begin{aligned}
\mathbf{e} &\implies \vec{v}_e = (x_1, x_2, \cdots, x_n), & \forall \mathbf{e} \in \mathcal{E}; \\
\mathbf{r} &\implies \vec{v}_r = (\delta_1, \delta_2, \cdots, \delta_n), & \forall \mathbf{r} \in \mathcal{R};
\end{aligned} \tag{22}$$

both of which are $n$-dim. Here both $x_i$ and $\delta_i$ are real numbers. In a triplet, the relation $\vec{v}_r$ acts as an addition vector added to the head entity $\mathbf{e}_1$. If one further chooses $L_p$-norm as the similarity measure:

$$\|(\vec{r} + \vec{e}_1) - \vec{e}_2\|_p, \tag{23}$$

this actually corresponds to the well-known **TransE** model (Bordes et al., 2013). There was a regularization in the original TransE mode that changes the entity rep-space, which however has been removed in many later works by properly bounding the negative scores.

## D.2  GROUP: $U(1)$

Use $n$-copies of $U(1)$, the 1-dim unitary transformation group, for the relational embedding. This is a ***compact abelian*** group. The simplest rep-space would be the real field $\mathbb{C}$, which should also appear $n$ times as $\mathbb{C}^n$. The group embedding then produces the following embedding vectors:

$$\begin{aligned} \mathbf{e} &\implies \vec{v}_e = (x_1, x_2, \cdots, x_n), && \forall \mathbf{e} \in \mathcal{E}; \\ \mathbf{r} &\implies \vec{v}_r = (\phi_1, \phi_2, \cdots, \phi_n), && \forall \mathbf{r} \in \mathcal{R}; \end{aligned} \tag{24}$$

where $x_i$ is a complex number containing a both real and imaginary part, while $\phi_i$ is a phase variable take values from 0 to $2\pi$. Therefore the entity-embedding dimension is $2n$, while the relation dimension is $n$. In a triplet, the relation $\vec{v}_r$ acts as a phase shift on the head entity $\mathbf{e}_1$. In a matrix form, one could define $\mathbf{R}_r$ as the diagonal matrix with the $i$-th diagonal element being $e^{i\phi_i}$. If one further chooses $L_p$-norm as the similarity measure:

$$\|\mathbf{R}_r \cdot \vec{v}_{e_1} - \vec{v}_{e_1}\|_p = \|[e^{i\vec{r}}] \circ \vec{e}_1 - \vec{e}_2\|_p, \tag{25}$$

where $\circ$ means a Hadamard product. This precisely leads to the **RotatE** model (Sun et al., 2019).

On the other hand, one could also use the $n$-torus $\mathbb{T}^n$ as the rep-space:

$$\begin{aligned} \mathbf{e} &\implies \vec{v}_e = (\theta_1, \theta_2, \cdots, \theta_n), && \forall \mathbf{e} \in \mathcal{E}; \\ \mathbf{r} &\implies \vec{v}_r = (\phi_1, \phi_2, \cdots, \phi_n), && \forall \mathbf{r} \in \mathcal{R}; \end{aligned} \tag{26}$$

where $\theta_i$ represents a coordinate on the torus. Still using the $L_p$-norm similarity measure:

$$\|\mathbf{R}_r \cdot \vec{v}_{e_1} - \vec{v}_{e_1}\|_p = \|e^{i\vec{r}} \circ e^{i\vec{e}_1} - e^{i\vec{e}_2}\|_p, \tag{27}$$

which leads to the **TorusE** model (Ebisu & Ichise, 2018) In the original implementation of TorusE, there is an additional projection $\pi$ from $\mathbb{R}^n$ to $\mathbb{T}^n$.

## D.3  GROUP: $SU(2)$

The 2D complex special unitary group $SU(2)$ is one of the simplest continuous nonabelian group, which is also the one we use in our instantiation. Use $n$-copies of $SU(2)$, the 2-dim special unitary group, for the relational embedding. This is a ***compact nonabelian*** group. The natural choice of entity rep-space would be $[\mathbb{C}^2]^{\otimes n}$, which consists of $n$-copies of $\mathbb{C}^2$. Each $\mathbb{C}^2$ subspace transforms as the standard rep-space of $SU(2)$. All relations thus act as $2n \times 2n$ block diagonal matrix, with each block being a $2 \times 2$ matrix carrying the standard representation of $SU(2)$. our group embedding is then fixed as:

$$\begin{aligned} \mathbf{e} &\implies \vec{v}_e = (x_1, y_1, \ x_2, y_2, \ \cdots, \ x_n, y_n), && \forall \mathbf{e} \in \mathcal{E}; \\ \mathbf{r} &\implies \vec{v}_r = (\alpha_1, \hat{\mathbf{n}}_1, \ \alpha_2, \hat{\mathbf{n}}_2, \ \cdots, \ \alpha_n, \hat{\mathbf{n}}_n), && \forall \mathbf{r} \in \mathcal{R}; \end{aligned} \tag{28}$$

where $x_i$ and $y_i$ are complex numbers, and $\alpha_i$ and $\hat{\mathbf{n}}_i = (\theta_i, \phi_i)$ represent angles. In a triplet, the relation $\vec{v}_r$ acts as a block diagonal matrix $\mathbf{R}_r$, with each $2 \times 2$ block $\mathbf{M}_i$ parametrized as:

$$\begin{bmatrix} \cos\alpha_i + i\sin\alpha_i\sin\theta_i & ie^{-i\phi_i} \cdot \sin\alpha_i\cos\theta_i \\ ie^{-i\phi_i} \cdot \sin\alpha_i\cos\theta_i & \cos\alpha_i - i\sin\alpha_i\sin\theta_i \end{bmatrix}$$

An operation of relation $\mathbf{r}$ on $\mathbf{e}$ would act each block matrix $M_i$ in the subspace of $(x_i, y_i)$. The complete model form after a $L_p$-norm is implemented would be:

$$\|\mathbf{R}_r \cdot \vec{v}_{e_1} - \vec{v}_{e_2}\|_p \tag{29}$$

## D.4   GROUP: $SO(3)$

The 3D rotation group $SO(3)$ is also one of the simplest continuous nonabelian group and has not been studied yet. A proper rep-space for entity embedding would be $[\mathbb{R}^3]^{\otimes n}$, which consists of $n$-copies of $\mathbb{R}^3$. Each $\mathbb{R}^3$ subspace transforms as the standard rep-space of $SO(3)$. All relations thus act as $3n \times 3n$ block diagonal matrix, with each block being a $3 \times 3$ matrix carrying the standard representation of $SO(3)$.

Next, we choose a proper parametrization of $SO(3)$. Instead of the more general angular momentum parametrization, due to our choice of using the standard representation, we could parameterize the $SO(3)$ elements using Euler angles $(\phi, \theta, \psi)$, which is easier for implementation.

Put all together, our group embedding is then fixed as:

$$\mathbf{e} \implies \vec{v}_e = \big(x_1, y_1, z_1, \ x_2, y_2, z_2, \ \cdots, \ x_n, y_n, z_n\big), \qquad \forall \mathbf{e} \in \mathcal{E};$$
$$\mathbf{r} \implies \vec{v}_r = \big(\phi_1, \theta_1, \psi_1, \ \phi_2, \theta_2, \psi_2, \ \cdots, \ \phi_n, \theta_n, \psi_n\big), \qquad \forall \mathbf{r} \in \mathcal{R}; \tag{30}$$

both of which are $3n$-dim. In a triplet, the relation vector $\vec{v}_r$ acts as a block diagonal matrix $\mathbf{R}_r$, with each $3 \times 3$ block $\mathbf{M}_i$ parametrized as following[6]:

$$\begin{bmatrix} \cos\psi_i\cos\phi_i - \cos\theta_i\sin\psi_i\sin\phi_i & \cos\psi_i\sin\phi_i + \cos\theta_i\cos\psi_i\cos\phi_i & \sin\psi_i\sin\theta_i \\ -\sin\psi_i\cos\phi_i - \cos\theta_i\sin\psi_i\cos\phi_i & -\sin\psi_i\sin\phi_i + \cos\theta_i\cos\psi_i\cos\phi_i & \cos\psi_i\sin\theta_i \\ \cos\psi_i\sin\theta_i & -\cos\psi_i\cos\theta_i & \cos\theta_i \end{bmatrix}$$

An operation of relation $\mathbf{r}$ on $\mathbf{e}$ would act each block matrix $M_i$ in the subspace of $(x_i, y_i, z_i)$. The complete model form after a $L_p$-norm is implemented would be:

$$\|\mathbf{R}_r \cdot \vec{v}_{e_1} - \vec{v}_{e_2}\|_p \tag{31}$$

## D.5   GROUP: $GL(m, \mathbb{V})$

This is a class of groups ($m$-dim general linear groups defined on field-$\mathbb{V}$). We separate the discussion into two parts according the commutativity property of the group:

1. **Group $GL(1, \mathbb{V})$:** Use $n$-copies of $GL(1, \mathbb{V})$, the 1-dim general linear group defined on field-$\mathbb{V}$, for the relational embedding. This is a ***noncompact abelian*** group. Suppose cos-similarity is used, and, most naturally, the entity rep-choice is chosen as $\mathbb{V}^n$. One has at least two simple choices for the field-$\mathbb{V}$: either real or complex.

   For the real field $\mathbb{R}$, embeddings are:

   $$\mathbf{e} \implies \vec{v}_e = \big(x_1, \ x_2, \ \cdots, \ x_n\big), \qquad \forall \mathbf{e} \in \mathcal{E};$$
   $$\mathbf{r} \implies \vec{v}_r = \big(s_1, \ s_2, \ \cdots, \ s_n\big), \qquad \forall \mathbf{r} \in \mathcal{R}; \tag{32}$$

   with both $x_i$ and $s_i$ being real numbers. In a triplet, the relation vector $\vec{v}_r$ acts as a diagonal matrix $\mathbf{R}_r$, with the $i$-th element being $s_i$. If one uses a cos-similarity which can be captured by the inner product, the eventual form would be:

   $$\vec{v}_{e_2}^T \cdot \big[\mathbf{R}_r \cdot \vec{v}_{e_1}\big] \tag{33}$$

   which leads to the **DisMult** model. And similarly a complex field $\mathbb{C}$ leads to a model structure similar to **ComplEx** (Trouillon et al., 2016). There is a minor difference in ComplEx: authors actually use the cos-similarity in a projected subspace (the real-subspace of $\mathbb{C}^n$):

   $$Re\left(\vec{v}_{e_2}^T \cdot \big[\mathbf{R}_r \cdot \vec{v}_{e_1}\big]\right) \tag{34}$$

2. **Group $GL(n, \mathbb{V})$:** Use $GL(n, \mathbb{V})$, the $n$-dim general linear group defined on field-$\mathbb{V}$, while keeping both the similarity measure (cos-similarity) and entity rep-space ($\mathbb{V}^n$) choices unchanged. This is a ***noncompact nonabelian*** group. The choice of $\mathbb{V} = \mathbb{R}$ would leads to a model similar to **RESCAL** (Nickel et al., 2011). However, different from the correspondences above, the original RESCAL model does not have a built-in group structure: it uses arbitrary $n \times n$ real matrices, some of which may not be invertible, and hence are not group elements in $GL(n, \mathbb{R})$. It is, therefore, worth to add the extra invertible constraint in RESCAL, which requests matrices constructed through group parameterization rather than assigned arbitrary matrix elements.

---

[6]See Appendix.E for a detailed explanation.

## D.6 GROUP: AFF($\mathbb{V}$)

Use Aff($\mathbb{V}$), the affine group defined on the field-$\mathbb{V}$, for the relational embedding. This is a ***noncompact nonabelian*** group. One could choose $\mathbb{V} = \mathbb{R}^n$ since the entity rep-space should always share the same one. If one further takes $L_p$-norm as the similarity measure, the embedding model would be quite similar to that in **TransR** (Lin et al., 2015) if relation and entity embeddings share the same dimension. However, similar to the RESCAL case, one important distinction is the original model does not explicit constraint the relational embedding on a group manifold. An extra invertible requirement would then produce exactly a group embedding model as proposed here.

## E EXPLANATION OF EULER ANGLES AND RELATION OPERATIONS

In this appendix section, we provide a pictorial explanation for Euler angles used in the exampling model of $SO(3)$.

The idea is that all 3D rotations that preserve orientation could be decomposed into the following 3 subsequent rotations:

$$\mathbf{D} = \begin{pmatrix} \cos\phi & \sin\phi & 0 \\ -\sin\phi & \cos\phi & 0 \\ 0 & 0 & 1 \end{pmatrix}, \tag{35}$$

$$\mathbf{C} = \begin{pmatrix} 1 & 0 & 0 \\ 0 & \cos\theta & \sin\theta \\ 0 & -\sin\theta & \cos\theta \end{pmatrix}, \tag{36}$$

$$\mathbf{B} = \begin{pmatrix} \cos\psi & \sin\psi & 0 \\ -\sin\psi & \cos\psi & 0 \\ 0 & 0 & 1 \end{pmatrix}. \tag{37}$$

The composed operation $\mathbf{M} = \mathbf{BCD}$ could then be written as:

$$\begin{bmatrix} \cos\psi\cos\phi - \cos\theta\sin\psi\sin\phi & \cos\psi\sin\phi + \cos\theta\cos\psi\cos\phi & \sin\psi\sin\theta \\ -\sin\psi\cos\phi - \cos\theta\sin\psi\cos\phi & -\sin\psi\sin\phi + \cos\theta\cos\psi\cos\phi & \cos\psi\sin\theta \\ \cos\psi\sin\theta & -\cos\psi\cos\theta & \cos\theta \end{bmatrix},$$

which is the matrix used in the main text. The meaning of the above three rotations can be easily visualized as following:

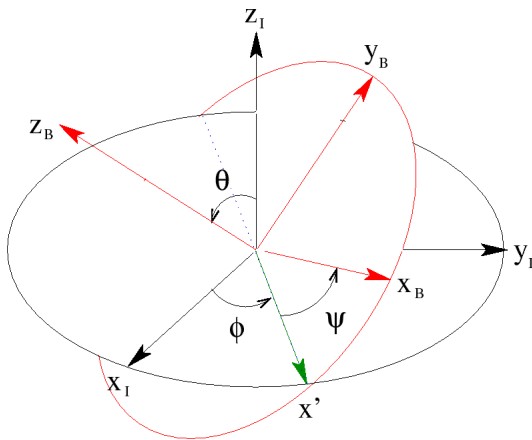

Figure 1: A visualization of Euler angles.

The action in the whole $\mathbb{R}^{3n}$ space looks like:

$$
\begin{bmatrix}
\mathbf{M}_1 & 0 & \dots & 0 \\
0 & \mathbf{M}_2 & \dots & 0 \\
\vdots & \vdots & \ddots & \vdots \\
0 & 0 & \dots & \mathbf{M}_n
\end{bmatrix}
\begin{bmatrix}
x_1 \\ y_1 \\ z_1 \\ x_2 \\ y_2 \\ z_2 \\ \vdots \\ x_n \\ y_n \\ z_n
\end{bmatrix}
\tag{38}
$$

