# OpenReview forum: "A Group-Theoretic Framework for Knowledge Graph Embedding"
_ICLR.cc/2020/Conference — Reject_

### Official Review · AnonReviewer2 · 2019-10-23
**Official Blind Review #2**

**Rating:** 3

**Review:**

I would like to thank the authors for the detailed rebuttal.

My concern is generalizing something that is not correctly describing the object we should model (KGs).

For instance, "all properties mentioned in the paper, including closure, identity, inversion, and associativity, are desired properties of relation embedding spaces, rather than existing properties of specific knowledge graphs. " => should we not care about the actual relations of knowledge graphs? Should we not build a theory that closely matches the properties of the actual objects we are modeling? The argument seems to be: forget the actual knowledge graph, here is a generalization of existing KGE methods. Even the proof provided was unrelated to my question: I gave an example: can you list all relations in FB15K and prove that they fit the group axioms? If they don't fit, what is the advantage of modeling the relations as a group? What is the representation error this causes?


The authors misunderstood my comment about (Bloem-Reddy and Teh 2019). "Therefore the previous works mentioned by the reviewer above are irrelevant." => this is a weird statement. Permutation invariance is relevant to all objects that are graphs. I recommend reading the long history relating graph models and exchangeability (Persi Diaconis has a good overview). I was just mentioning that I don't see why KGs should have another associated permutation group besides permutation invariance. I would also recommend following the emerging literature in graph representation learning using group theory, of which KGs are but a special case.

"The (entity) manifold, however, means that the relations no longer form a group. I see no easy fix." => I clarified it, since I was talking about the entity embedding.

I did not see a revised version of the manuscript.


I will raise my score because the paper could be published as a niche paper. We generalize KGE but we acknowledge its shortcomings: And here is a way to measure the embedding error of modeling the relations as a group, when they are not actually a group.

--------------

The paper is well written and an interesting read. I think it complements well the existing literature.

Unfortunately, I think the paper overstate its claims. It is clear that permutation groups are the natural language of all graphs (Kondor and Trivedi, 2018) and (Bloem-Reddy and Teh, 2019). It is less clear that knowledge graphs also comply with another set of group axioms. Why would the set of relations be closed? Say, Owns \cdot Spouse_of = ? . I cannot see why and how this is necessary. Must all the relations be closed? I understand why this is true for KGE but this is not true for KGs, which only shows that KGEs may not be the right method to represent KGs moving forward. I see no easy fix.

The rest of the paper is straightforward, just applying the definitions of groups. I found the classification of different methods interesting, and should be made more clear in the experiments.

The experimental results are interesting but their significance is unclear. Please add standard deviations to all experiments. We cannot have a sense of the significance of the results without knowing how many runs were executed, how they were executed (e.g., k-fold cross-validation, bootstrapping), and their standard deviation.

I dispute that the paper proves that the definition of groups emerges purely from the nature of the knowledge graph embedding (KGE) task. I would actually say the opposite, it is clear to me that for most KGs, their relations do not fit the definition of a group. I am open to be (formally) proven wrong. E.g., can you list all relations in FB15K and show that they fit the group axioms?

Fixing the paper: Maybe the paper could be rewritten, constrained to Euclidean spaces? Then prove (formally) that the group axioms are a sufficient(?) and necessary(?) condition for such embeddings?

Minor issues:
honestly could not make sense of the abstract:
The sentence choices
-  "which suggests that a group-based embedding framework is essential for model design"
- "Our theoretical analysis explores merely the intrinsic property of the embedding problem itself without introducing extra design"
in the abstract are very strange. It means nothing to a reader at that point. What a model design? What is an extra design? A modeling assumption? A model prior?

"Using the proposed framework, one could construct embedding models that naturally accommodate all possible local graph patterns, which are necessary for reproducing a complete graph from atomic knowledge triplets" => what is an "atomic knowledge triplet"? Why are they necessary to reproduce a complete graph? This is all very confusing.

" contradicts to the entity regularization" =>  contradicts the entity regularization

References:
Kondor, R. and Trivedi, S., 2018. On the generalization of equivariance and convolution in neural networks to the action of compact groups. ICML
Bloem-Reddy, B. and Teh, Y.W., 2019. Probabilistic symmetry and invariant neural networks. arXiv:1901.06082.


**Experience Assessment:**

I have published one or two papers in this area.

**Review Assessment: Checking Correctness Of Derivations And Theory:**

I carefully checked the derivations and theory.

**Review Assessment: Checking Correctness Of Experiments:**

I assessed the sensibility of the experiments.

**Review Assessment: Thoroughness In Paper Reading:**

I read the paper thoroughly.

---

> ### Author Response · Authors · 2019-11-10
> **response-1**
>
> We thank the reviewer for the detailed review and would like to address the following points:
>
>
> 1.
> Reviewer: “Why would the set of relations be closed? Say, Owns \cdot Spouse_of = ? I cannot see why and how this is necessary. Must all embedding methods be closed? I dispute that the paper proves that the definition of groups emerges purely from the nature of the knowledge graph embedding (KGE) task. I would actually say the opposite, it is clear to me that for most KGs, their relations do not fit the definition of a group. I am open to be (formally) proven wrong. E.g., can you list all relations in FB15K and show that they fit the group axioms?”
>
> Response:
> This concern is related to an important difference between the following two things: the relation pattern in a specific KG and the structure of the relation embedding space. We have discussed this in a very careful way in Sec.3.2 of our manuscript. To put it in short: all properties mentioned in the paper, including closure, identity, inversion, and associativity, are desired properties of relation embedding spaces, rather than existing properties of specific knowledge graphs.
>
> In our paper, we choose the proper words very carefully to address this difference. For the closure property, we phrased as: “To allow the possible existence of composition, in general, the elements R1 · R2 should also be an element living in the same relation-embedding space”, which clearly suggests that it is a requirement for the *relation-embedding space* to accommodate *possible existence of composition* in specific KG datasets. Besides, we also explained explicitly, in the footnote on page.3, that “Given a graph, not all compositions correspond to meaningful relations, but an embedding model should be able to capture this possibility in general”.
>
> ================================
> Below we give a more detailed discussion:
>
> a. The relation patterns in a specific KG:
> We completely agree that for a specific KG, the meaningful relations existing in the graph may not form an exact group. However, it is clear that some relation patterns indeed can emerge from compositions and inversions in real KGs, which is the exact concern taken by a lot of proceeding studies, including RotatE, QuatE, and DihEdral. The problem is that, in a specific KG, these patterns, which we denote as super-relations, are not accessible to users/researchers, since it is not practical at all to enumerate every single pattern in a large KG.
>
> b. The structure of the relation embedding space:
> When proposing a model for general KGE tasks, due to the difficulty of directly access all super-relations in every single task, one should try to accommodate all possibilities. Take the ‘closure’ as an example, in a specific KG dataset, one might not know if certain compositional relations are existing. Note that there are two types of‘ compositions’: conceptual and mathematical ones. The conceptual one depends on the specific dataset; the mathematical one depends on the embedding method. To accommodate the possible existence of the conceptual composition r1· r2, whatever representations, say R1 and R2, are assigned to r1 and r2, the mathematical composition R1· R2 must also be contained in the relation embedding space. For instance, in RotatE, where each relation is a U(1) rotation, the composition of two U(1) elements are still contained in the U(1) group.
>
> Prove the requirement of closure by contradiction: Suppose abstract relation r1, r2 are embedded as R1, R2. Importantly, since the operation rule of a relation-embedding R and an entity-embedding E are already fixed, the mathematical composition rule between R1 and R2 has also been fixed: it is defined as operating by R1 first then by R2 subsequently. If the closure is absent (i.e. R1· R2 is not contained in the embedding space), this method would never be able to correctly represent the r1· r2. Therefore, the model would fail the KGE tasks where r1· r2 exists.
> ================================

---

> ### Author Response · Authors · 2019-11-10
> **response-2**
>
> 2.
> Reviewer: It is clear that permutation groups are the natural language of all graphs (Kondor and Trivedi, 2018) and (Bloem-Reddy and Teh, 2019)
>
> Response: There is a huge difference between the works on equivariance as mentioned by the reviewer and our current work. Formally, these two sets of work consider difference group action spaces.
>
> In previous works (Kondor and Trivedi 2018, Bloem-Reddy and Teh 2019), each basic unit (a vector in vector spaces, or a point in topological spaces) in the group action spaces represent a whole graph/image. A group action transforms one graph to another graph. While in our current work, each basic unit represents one entity (i.e., a node in a graph). A group action transforms one node to another node. Therefore the previous works mentioned by the reviewer above are irrelevant. We would add the following comments to our manuscript:
>
> a. In the first set of problems, permutations describe pixel transformations and provide a nice description if the considered actions are discrete. The analyses can’t be straightforwardly applied to continuous groups: although a permutation group can assist the representation theory of Lie groups via Schur-Weyl duality, it requires a completely different analysis.
>
> b. It is possible abstractly represent any group action as a permutation. However, in the case of KGE problems, this does not provide any help for embedding-model design. To use permutation description, one has to construct a tabular mapping from entity to entity, which requires much more computational resources and does not offer any advantage in practice.
>
>
> 3.
> Reviewer: I understand it could be the case for Euclidean embeddings, which only showcases that Euclidean embeddings are not ideal for KGs. But can't embeddings be defined over a (Riemann?) manifold, such that some embedding mappings (Owns \cdot Spouse_of = ? ) need not be defined?
>
> Response: It seems the reviewer is considering the entity embedding (correct us if we are wrong). However, our group manifold (or say the parameter space) is the relation embedding space.
>
> It is quite clear that our above discussion does not depend on any modeling details. It starts with analyzing a KGE embedding task, then produces requirements for a general embedding model. The four required properties: closure, identity, associativity, and inverse, are defined purely algebraically and has nothing to do with a geometric viewpoint of groups. We cannot see the Euclidean metric would play any role here: the existence of (r1 · r2) is determined by the KG dataset.
>
>
> 4.
> Reviewer: The manifold, however, means that the relations no longer form a group. I see no easy fix. Btw, I disagree with what TorusE did (the relations described in the paper do not form a Lie group).
>
> Response: It is well-known that all compact Lie groups correspond to manifolds which are non-Euclidean (since it’s compact): Torus is a compact manifold (non-Euclidean), which is a Lie group; S^3 is a compact manifold (non-Euclidean), which is also a Lie group. Hence, we see no need for a fix. TorusE never embeds relations as a group. Instead, it embeds entities on a torus, which forms a Lie group manifold.
>
>
> 5.
> Reviewer: “The rest of the paper is straightforward, just applying the definitions of groups.”
>
> Response: More precisely, what we applied in the later sections are not definitions of groups, but representations of groups. For example, in the SU(2) case we applied the 2-dim complex representation; and in the SO(3) case we applied a 3-dim real representation.
>
>
> 6.
> Reviewer: “The experimental results are interesting but their significance is unclear. Please add standard deviations to all experiments. We cannot have a sense of the significance of the results without knowing how many runs were executed, how they were executed (e.g., k-fold cross-validation, bootstrapping), and their standard deviation.
>
> Response: Our experiments are reported as a 5-run average, we will add more information about this experiment data in the revised version.
>
>
> 7.
> Reviewer: Minor issues.
>
> Response: We thank the reviewer for pointing out minor issues, we will rephrase in the revised version.

---

### Official Review · AnonReviewer1 · 2019-10-23
**Official Blind Review #1**

**Rating:** 3

**Review:**

This paper start merely by studying the graph reconstruction problem and prove that the intrinsic structure of this task itself automatically produces the complete definition of groups. it seems to be a novel result. Based on this result, one could construct embedding models that naturally accommodate all possible local graph patterns, and the paper also shows a few simulations.

My main concern is that, while the focus on this work is the theoretical finding, there is no rigorous statement of it as a theorem. As a result, I am not exactly sure what the proofs in the appendix is trying to show. In addition, the proofs seems to be very trivial.

For the algorithm section, I feel that it is also lacking in the sense that there is still no automatic way to choose which group to embed. It is also unclear what is the purpose of the simulation section. While it says "As theoretically analyzed in Section 3.2, and empirically shown above, continuous nonabelian groups are more reasonable choices for general tasks", the advantage of continuous nonabelian groups are not so significant in the tables.

**Experience Assessment:**

I do not know much about this area.

**Review Assessment: Checking Correctness Of Derivations And Theory:**

I did not assess the derivations or theory.

**Review Assessment: Checking Correctness Of Experiments:**

I did not assess the experiments.

**Review Assessment: Thoroughness In Paper Reading:**

I read the paper at least twice and used my best judgement in assessing the paper.

---

> ### Author Response · Authors · 2019-11-10
> **response**
>
> We thank the reviewer for the detailed review and would like to address the following points:
>
>
> 1.
> Reviewer:  “there is no rigorous statement of it as a theorem. As a result, I am not exactly sure what the proofs in the appendix is trying to show. In addition, the proofs seems to be very trivial.”
>
> Response: This work does not aim at proving a ‘theorem’. Instead, it is more about a discovery: our theoretical analysis discovers an intrinsic structure of general knowledge graph embedding tasks, which coincides with the algebraic definition of mathematical groups. In other words, the theoretical analysis is trying to show the emergence of the mathematical object, group, from the requirements of a general KGE problem. It should not matter too much whether or not the analysis is formulated into a theorem. If one insists, though, the main finding mentioned above can be formulated as the following theorem:
> “Theorem: to accommodate the most general knowledge graph structures, the relation embedding space would form a group parameter space (or group manifold).”
>
> The proof in the appendix is provided to show that relations in a knowledge graph, which are naturally defined as mappings from one entity to another, obey the ‘associativity’ property. That is, for a chain of mappings, we always have:  (r1 · r2) · r3 = r1 · (r2 · r3). Although the mathematics of the proof in the appendix seems trivial, it is important and significant because it can hold if and only if a proper definition of composition law is rigorously provided. This is achieved in our case: the operation of the composed relation (r1 · r2)  is defined as two ordered sequential operations by r1 and r2. Note that this definition of composition law is reasonable and arise naturally from relation mapping in KGE tasks, rather than an artificial fact or hypothesis.
>
> To summarize, the key contribution of this work is to give a rigorous *mathematical formulation* of KGE tasks themselves.
>
>
> 2.
> Reviewer: “For the algorithm section, I feel that it is also lacking in the sense that there is still no automatic way to choose which group to embed. It is also unclear what is the purpose of the simulation section. While it says "As theoretically analyzed in Section 3.2, and empirically shown above, continuous nonabelian groups are more reasonable choices for general tasks", the advantage of continuous nonabelian groups are not so significant in the tables.”
>
> Response: To solve a generic KGE task, there are three stages:
> 1. What are the task’s requirements for the model?
> 2. What type of models can satisfy these requirements?
> 3. How to construct a specific model for practical usage?
>
> For the first question, our analysis in Sec.3  finds the requirements that coincide with the definition of groups (this is the first work that formally proved this as we know); therefore, for the second one, group manifolds are natural choice for relational embedding model; for the third one, we have provided a general recipe to automatically construct a model as long as the embedding group is chosen.
>
> In this sense, our work targets at the most general KGE tasks, rather than on a specific one. The reviewer’s concern about “which group to embed” is asking a detailed version of the second question. General KGE problems only restrict the embedding space from arbitrary spaces into group manifolds, while “which group is proper” depends on the details of specific tasks.
>
> Besides, with the most general concerns, we explained that the choice of groups could be further narrowed into the category of *continuous* *non-abelian* groups. Continuous groups are more efficient for gradient-based tasks, and the non-abelian nature could handle relation compositions which are not commutative, both of which concern general KGE problems.
>
> While these are all the restrictions one could derive for generic tasks, in practice, if more details of the given task are available, one could further restrict the choice of groups into a smaller category of groups. For example, if one knows in advance that all (or most) relation compositions are commutative in a specific task, then a larger non-abelian group would be redundant, and one could simply use an abelian sub-group, which is much smaller and computationally efficient.

---

### Official Review · AnonReviewer3 · 2019-10-29
**Official Blind Review #3**

**Rating:** 3

**Review:**

The authors approach the problem of representing knowledge graphs from a top-down perspective, in which they argue for certain fundamental desiderata for hyper-relations (composition, inversion) followed by properties that define a mathematical group.

I found the paper extremely difficult to follow. As defined in Eq. 1, knowledge graph embeddings are a model family with a choice of domain for the entity and relation, and a choice of how that relation operates on a head entity. This means one can devise arbitrary properties and restrictions on that family. It's not clear to me what motivates selecting a (abelian) group, where inversion, closure, identity, associativity, and commutativity are demanded to be properties of knowledge graph embedding models. This seems more a definition of what models they consider, rather than a novel insight about knowledge graph embedding models itself (the authors claim "we proved for the first time the emergence of a group definition in the KG representation learning", which seems hard to wrap one head's around).

Given this overarching family of models, the authors proceed to identify existing models as certain choices of that family.  I see little use in inventing this abstraction as the authors do not show any practical insights, or interesting theoretical analysis that comes from this higher-level abstraction.

**Experience Assessment:**

I do not know much about this area.

**Review Assessment: Checking Correctness Of Derivations And Theory:**

I assessed the sensibility of the derivations and theory.

**Review Assessment: Checking Correctness Of Experiments:**

I assessed the sensibility of the experiments.

**Review Assessment: Thoroughness In Paper Reading:**

I read the paper at least twice and used my best judgement in assessing the paper.

---

> ### Author Response · Authors · 2019-11-10
> **response**
>
> We thank the reviewer for the review and would like to address the following points:
>
>
> 1.
> Reviewer: “It's not clear to me what motivates selecting a (abelian) group, where inversion, closure, identity, associativity, and commutativity are demanded to be properties of knowledge graph embedding models. This seems more a definition of what models they consider, rather than a novel insight about knowledge graph embedding models itself (the authors claim "we proved for the first time the emergence of a group definition in the KG representation learning", which seems hard to wrap one head's around).”
>
> Response: It seems that the reviewer failed to grasp the delivery in our paper, we did not recommend the Abelian group at all — instead, with all the analysis on a general KGE task in Sec.3 of the paper, we suggest *non-Abelian* groups for relation embeddings.
>
> It is not clear what the reviewers mean by “a definition of what models they consider” since what Eq.1 describes is a general knowledge graph, not a model at all. And, importantly, the four properties: closure, identity, inverse, and associativity (along with non-commutativity), are the most general requirements of KGE tasks themselves, not restricted to any specific model structures.
>
>
> 2.
> Reviewer: “Given this overarching family of models, the authors proceed to identify existing models as certain choices of that family.  I see little use in inventing this abstraction as the authors do not show any practical insights or interesting theoretical analysis that comes from this higher-level abstraction.”
>
> Response: We do not understand what “family” the reviewer is referring to. We would like to emphasize again: the properties in Sec.3.2, which happen to coincide with the mathematical definition of groups, are requirements of general KGE tasks, not restricted to any specific modeling structures. We offer a deep understanding of these requirements, following which, a natural choice for general KGE model would be using non-Abelian continuous groups. Our recipe in Sec.4 describes in detail how to construct the embedding model when a group is selected, which directly produces models for *practical usage*. Our experimental instantiation selected SU(2) as the embedding group and constructed the corresponding embedding model that can be used in practice, which finished a complete demonstration of the proposed framework. Without this understanding, there is no general recipe to systematically construct KGE models at all.

---

### Public Comment · ~Chen_Cai1 · 2019-10-04
**Relevant paper**

Hello,

It is very interesting to see that you have a very similar idea about introducing group theory into KGE. I also wrote a small paper connecting group representation theory with KGE, which is recently accepted in NeurIPS graph representation workshop.

Best,
Chen

Group Representation Theory for Knowledge Graph Embedding
https://grlearning.github.io/papers/15.pdf
Another relevant paper (NeurIPS 2019) is Quaternion Knowledge Graph Embeddings https://arxiv.org/pdf/1904.10281.pdf

---

> ### Author Response · Authors · 2019-10-06
> **response**
>
> We appreciate that the reviewer pointed out two relevant works [1, 4]. Since they are still under proceeding, they slipped out attention. We will cite them in our manuscript. Below, we discuss the major differences between our work and them.
>
> The work in [1] nicely introduced the mathematical definition of groups along with related important mathematical concepts. With such concepts, the authors [1] provide an alternative theoretical explanation for RotatE [2] in the field of abelian group embeddings, which is a subclass in our proposed group embedding framework. In our manuscript, we cited DihEdral [2] (see Section 2 on comparing related works) that provided a gentle introduction to groups. [1] enriches the discussion with more mathematical ingredients including representation theory, Schur’s Lemma, etc.
>
> We would like to emphasize a major contribution in our work, compared with both [1] and [3]. Our theory does not aim at *introducing* a group perspective. Instead, we *proved* the existence of the definition of groups, merged purely from the nature of the knowledge graph embedding (KGE) task. Following our findings, group theory becomes the essential language for the problem rather than simply an alternative perspective for improving KGE models. That is our theory lays down a unified theoretical foundation for the KGE research.
>
> The work in [4] mentioned by the reviewer is very interesting. The authors proposed to use quaternions (also octonions and sedenions in the appendix), which served as an extension of complex numbers, to improve performance. The intuition of their model was not initiated from group theory. However, in Model Analysis (Section 5.3 in [4]), the authors stated that:
>
>         “Normalizing the relation to unit quaternion is a critical step for the embedding performance. This is likely because scaling effects in non-unit quaternions are detrimental.”
>
> However, an explicit reason is absent. After a thorough investigation, we realized quite an interesting fact: there is a mathematical correspondence between unit quaternions and SU(2) group used in our work. More rigorously, SU(2) group is an isomorphism [5] of unit quaternions. This explains the necessity of applying unit quaternions: only unit quaternions are consistent with the group structure, while non-unit ones cannot. The QuatE model [4] embeds relations with unit quaternions, each of which can be mapped to a SU(2) matrix element in our work. Therefore, one could construct a one-to-one mapping, from the exampling model SU2E in our work to QuatE model in [4]. This indicates that QuatE model falls into the category of continuous non-Abelian group embedding (Section 3.3 Table 1 in our work) thus has the potential to outperform simpler groups (e.g. group U(1) in RotatE [2]).  Indeed, QuetE model achieved great performance on Freebase and WordNet experiments. We believe some of the performance boost might be rooted in the implementation details (including negative sampling and self-advisory approach, regularizations, value function forms and other hyperparameter setups), which is out of the scope of our work. (Our implementation codebase is derived from the public repository of [2].) Although we do not have an opportunity to replicate the result due to the recency of the work in [4] and not able to access to its codes, the performance advantage of QuatE model is in line with our proposed framework. It would be really interesting to adopt the QuatE implementation on other promising groups in the future.
>
> Finally, we want to highlight that the SU2E model serves only as an instantiation of our proposed modeling framework, which provides a generic recipe for constructing KGE models. This exhibits instructive guidelines for designing more general KGE models in future works and may motivate further theoretical analysis based on group theory.
>
>
> [1] Group Representation Theory for Knowledge Graph Embedding, Neurips workshop 2019
> [2] RotatE: Knowledge Graph Embedding by Relational Rotation In Complex Space, ICLR 2019
> [3] Relation Embedding with Dihedral Group in Knowledge Graph, ACL 2019
> [4] Quaternion Knowledge Graph Embeddings, Neurips 2019
> [5] https://en.wikipedia.org/wiki/Quaternion

---

> > ### Public Comment · ~Chen_Cai1 · 2019-10-07
> > **Some further comments**
> >
> > Thanks for the detailed reply! I also realized that the method proposed in the paper is essentially the same as Quaterninon embedding. It's quite interesting to see that the same model is discovered twice from different perspectives.

---

> > > ### Author Response · Authors · 2019-10-08
> > > **Clarification**
> > >
> > > 1. Our paper is delivering a systematic Group Embedding Framework, rather than a single model example. This framework is not equal to Quaternion Embedding (i.e. QuatE) [5] at all. Instead, QuatE [5], and equivalently our SU2E model, only serves as a demonstrating example of the framework; other examples include RotatE [1], TorusE [2], and DihEdral [3]. And in fact, we are implementing various other groups for KGE problems in our proceeding works right now.
> > >
> > > 2. We don't ‘discover’ models by, for example, testing different number systems (complex, quaternion, octonion, sedenion) as in ComplEx [4], RotatE [1]  and QuatE [5]. Following our framework, we can systematically construct them with a clear awareness of their expressive power. And in this sense, the work QuatE, in fact, demonstrates the power of continuous non-Abelian groups nicely.
> > >
> > >
> > > [1] RotatE: Knowledge Graph Embedding by Relational Rotation In Complex Space, ICLR 2019
> > > [2] TorusE: Knowledge graph embedding on a lie group. AAAI 2018
> > > [3] Relation Embedding with Dihedral Group in Knowledge Graph, ACL 2019
> > > [4] Complex embeddings for simple link prediction. ICML 2016
> > > [5] Quaternion Knowledge Graph Embeddings, Neurips 2019

---

> > > ### Public Comment · ~Hung-Nghiep_Tran1 · 2019-12-30
> > > **Another Quaternion-based knowledge graph embedding paper**
> > >
> > > I would like to add that an earlier Quaternion-based model was also discovered in [1] from the perspective of weighted sum of multi-embedding trilinear products, which is the first one to my knowledge. We proposed a simple model using the Quaternion algebra (Sect. 3.4) and showed promising results, outperforming ComplEx on comparable settings (Sect. 6.3).
> > >
> > > The results were published in an EDBT/ICDT workshop early 2019, but we did not follow this idea further because we were busy with a more advanced generalized idea and quite happy with it. We are glad to see that the NeurIPS paper nicely presents and confirms the advantage of Quaternion-based embedding, although there are still some rooms for improvement.
> > >
> > > [1] Analyzing Knowledge Graph Embedding Methods from a Multi-Embedding Interaction Perspective, https://arxiv.org/abs/1903.11406

---

> > ### Author Response · Authors · 2019-10-09
> > **Supplementary discussion**
> >
> > In the above discussion, we explained the “unit quaternion puzzle”: unit-quaternions work much better than non-unit ones in relation-embeddings, as only unit ones are consistent with the group structure of SU(2). Now we provide an explanation for another confusing while important phenomenon in [1].
> >
> > In the work of QuatE [1], there are in total three different number-systems mentioned: quaternions, octonions, and sedenions, all of which are extended from complex numbers. The quaternion based model QuatE has achieved great performance, while the octonion based model, namely, OctonionE, did not show any further promising improvement (Appendix 7.3 in [1]). Viewed from the number-system perspective, octonions (also sedenions) indeed should have been more expressive than quaternions. This is not consistent with the experimental results.
> >
> > We now interpret these two models from our Graph Embedding Framework perspective. The QuatE model, as discussed in detail above, can be related to the SU(2) group embedding model; at the same time, the OctonionE can also be related to a group embedding model using Spin(8), although the relation mapping would be more complicated. Limiting to a discussion on the three major properties of a group: continuousness, commutativity, and compactness, there is no essential difference between SU(2) and Spin(8). Both models fall into the category of continuous non-Abelian group embeddings, hence, should process similar expressive power. This explains why there is no further improvement in OctonionE, when compared with QuatE.
> >
> >
> > [1] Quaternion Knowledge Graph Embeddings, Neurips 2019

---

### Public Comment · ~Dai_Quoc_Nguyen1 · 2019-10-09
**Cite the ConvKB paper or simply do not include the TransE results on WN18RR and FB15K-237**

Hi,

I saw that the TransE results on WN18RR and FB15k-237 in your paper are taken from the ConvKB paper [1] without citing it. It would be nice if you can cite the ConvKB paper. Otherwise, you simply do not include TransE as a baseline on these two datasets.

Best,
Dai.

[1] A Novel Embedding Model for Knowledge Base Completion Based on Convolutional Neural Network. NAACL 2018.

---

> ### Author Response · Authors · 2019-10-09
> **response**
>
> Thanks for pointing out the missing citation. We would definitely include it in the revision.

---

### Decision · Program_Chairs · 2019-12-19

**Decision:**

Reject

**Comment:**

This paper presents a rigorous mathematical framework for knowledge graph embedding. The paper received 3 reviews. R1 recommends Weak Reject based on concerns about the contributions of the paper; the authors, in their response, indicate that R1 may have been confused about what the contributions were meant to be. R2 initially recommended Reject, based on concerns that the paper was overselling its claims, and on the clarity and quality of writing. After the author response, R2 raised their score to Weak Reject but still felt that their main concerns had gone unanswered, and in particular that the authors seemed unwilling to tone down their claims. R3 recommends Weak Reject, indicating that they found the paper difficult to follow and gave some specific technical concerns. The authors, in their response, express confusion about R3's comments and suggest that R3 also did not understand the paper. However, in light of these unanimous Weak Reject reviews, we cannot recommend acceptance at this time.  We understand that the authors may feel that some reviewers did not properly understand or appreciate the contribution, but all three reviewers are researchers working at highly-ranked institutions and thus are fairly representative of the attendees of ICLR; we hope that their points of confusion and concern, as reflected in their reviews, will help authors to clarify a revision of the paper for another venue.